# Serotype Distribution of Remaining Pneumococcal Meningitis in the Mature PCV10/13 Period: Findings from the PSERENADE Project

**DOI:** 10.3390/microorganisms9040738

**Published:** 2021-04-01

**Authors:** Maria Garcia Quesada, Yangyupei Yang, Julia C. Bennett, Kyla Hayford, Scott L. Zeger, Daniel R. Feikin, Meagan E. Peterson, Adam L. Cohen, Samanta C. G. Almeida, Krow Ampofo, Michelle Ang, Naor Bar-Zeev, Michael G. Bruce, Romina Camilli, Grettel Chanto Chacón, Pilar Ciruela, Cheryl Cohen, Mary Corcoran, Ron Dagan, Philippe De Wals, Stefanie Desmet, Idrissa Diawara, Ryan Gierke, Marcela Guevara, Laura L. Hammitt, Markus Hilty, Pak-Leung Ho, Sanjay Jayasinghe, Jackie Kleynhans, Karl G. Kristinsson, Shamez N. Ladhani, Allison McGeer, Jason M. Mwenda, J. Pekka Nuorti, Kazunori Oishi, Leah J. Ricketson, Juan Carlos Sanz, Larisa Savrasova, Lena Petrova Setchanova, Andrew Smith, Palle Valentiner-Branth, Maria Teresa Valenzuela, Mark van der Linden, Nina M. van Sorge, Emmanuelle Varon, Brita A. Winje, Inci Yildirim, Jonathan Zintgraff, Maria Deloria Knoll

**Affiliations:** 1Johns Hopkins Bloomberg School of Public Health, Baltimore, MD 21205, USA; yyang165@jhmi.edu (Y.Y.); jbenne63@jhu.edu (J.C.B.); kylahayford@jhu.edu (K.H.); sz@jhu.edu (S.L.Z.); meaganepeterson@gmail.com (M.E.P.); nbarzee1@jhu.edu (N.B.-Z.); lhammitt@jhu.edu (L.L.H.); 2Independent Consultant, 1296 Coppet, Switzerland; drf3217@gmail.com; 3World Health Organization, 1202 Geneva, Switzerland; dvj1@cdc.gov; 4Center of Bacteriology, National Laboratory for Meningitis and Pneumococcal Infections, Institute Adolfo Lutz (IAL), São Paulo 01246-902, Brazil; samanta.almeida@ial.sp.gov.br; 5Department of Pediatrics, Division of Pediatric Infectious Diseases, University of Utah Health Sciences Center, Salt Lake City, UT 84132, USA; Krow.Ampofo@hsc.utah.edu; 6National Centre for Infectious Diseases, National Public Health Laboratory, Singapore 308442, Singapore; michelle_lt_ang@ncid.sg; 7Malawi-Liverpool-Wellcome Trust Clinical Research Programme, P.O. Box 30096, Chichiri, Blantyre 3, Malawi; 8National Center for Emerging and Zoonotic Infectious Diseases, Centers for Disease Control and Prevention, Arctic Investigations Program, Division of Preparedness and Emerging Infections, Anchorage, AK 99508, USA; zwa8@cdc.gov; 9Department of Infectious Diseases, Italian National Institute of Health (Istituto Superiore di Sanità, ISS), 00161 Rome, Italy; romina.camilli@iss.it; 10Instituto Costarricense de Investigación y Enseñanza en Nutrición y Salud, Tres Ríos, 30301 Cartago, Costa Rica; gchanto@inciensa.sa.cr; 11CIBER Epidemiología y Salud Pública, (CIBERESP), 28029 Madrid, Spain; pilar.ciruela@gencat.cat (P.C.); mp.guevara.eslava@navarra.es (M.G.); 12Surveillance and Public Health Emergency Response, Public Health Agency of Catalonia, 08005 Barcelona, Spain; 13Centre for Respiratory Diseases and Meningitis, National Institute for Communicable Diseases of the National Health Laboratory Service, 2192 Johannesburg, South Africa; cherylc@nicd.ac.za (C.C.); JackieL@nicd.ac.za (J.K.); 14School of Public Health, Faculty of Health Sciences, University of the Witwatersrand, 2000 Johannesburg, South Africa; 15Irish Meningitis and Sepsis Reference Laboratory, Children’s Health Ireland at Temple Street, Temple Street, D01 YC76 Dublin 1, Ireland; mary.corcoran@cuh.ie; 16Distinguished Professor of Pediatrics and Infectious Diseases, The Faculty of Health Sciences, Ben-Gurion University of the Negev, Beer-Sheva, Israel; rdagan@bgu.ac.il; 17Department of Social and Preventive Medicine, Laval University, Québec, QC G1V 0A6, Canada; philippe.dewals@criucpq.ulaval.ca; 18Department of Microbiology, Immunology and Transplantation, KU Leuven, 3000 Leuven, Belgium; stefanie.desmet@uzleuven.be; 19National Reference Centre for Streptococcus Pneumoniae, University Hospitals Leuven, 3000 Leuven, Belgium; 20Faculty of Sciences and Health Techniques, Mohammed VI University of Health Sciences (UM6SS) of Casablanca, 20250 Casablanca, Morocco; idiawara@um6ss.ma; 21National Reference Laboratory, Mohammed VI University of Health Sciences (UM6SS), 82403 Casablanca, Morocco; 22National Center for Immunizations and Respiratory Diseases, Centers for Disease Control and Prevention, Atlanta, GA 30333, USA; ipe3@cdc.gov; 23Instituto de Salud Pública de Navarra—IdiSNA, 31003 Pamplona, Spain; 24Swiss National Reference Centre for Invasive Pneumococci, Institute for Infectious Diseases, University of Bern, 3012 Bern, Switzerland; Markus.Hilty@ifik.unibe.ch; 25Department of Microbiology and Carol Yu Centre for Infection, Queen Mary Hospital, The University of Hong Kong, Hong Kong, China; plho@hku.hk; 26National Centre for Immunisation Research and Surveillance and Discipline of Child and Adolescent Health, Faculty of Medicine and Health, Children’s Hospital Westmead Clinical School, University of Sydney, Westmead, NSW 2145, Australia; sanjay.jayasinghe@health.nsw.gov.au; 27Department of Clinical Microbiology, Landspitali—The National University Hospital, Hringbraut, 101 Reykjavik, Iceland; karl@landspitali.is; 28Immunisation and Countermeasures Division, Public Health England, London NW9 5EQ, UK; shamez.ladhani@phe.gov.uk; 29Toronto Invasive Bacterial Diseases Network, and Department of Laboratory, Medicine and Pathobiology, University of Toronto, Toronto, ON M5S 1A8, Canada; Allison.McGeer@sinaihealth.ca; 30World Health Organization Regional Office for Africa, P.O. Box 06, Brazzaville, Congo; mwendaj@who.int; 31Department of Health Security, Finnish Institute for Health and Welfare, 00271 Helsinki, Finland; pekka.nuorti@tuni.fi; 32Health Sciences Unit, Faculty of Social Sciences, Tampere University, 33100 Tampere, Finland; 33Toyama Institute of Health, Imizu, Toyama 939-0363, Japan; toyamaeiken1@chic.ocn.ne.jp; 34Department of Pediatrics, University of Calgary, Calgary, AB T3B 6A8, Canada; ljricket@ucalgary.ca; 35Laboratorio Regional de Salud Pública, Dirección General de Salud Pública, Comunidad de Madrid, 28053 Madrid, Spain; juan.sanz@salud.madrid.org; 36Centre for Disease Prevention and Control of Latvia, 1005 Riga, Latvia; larisa.savrasova@spkc.gov.lv; 37Doctoral Studies Department, Riga Stradinš University, 1007 Riga, Latvia; 38Department of Medical Microbiology, Faculty of Medicine, Medical University of Sofia, 1431 Sofia, Bulgaria; lenasetchanova@hotmail.com; 39Bacterial Respiratory Infection Service, Scottish Microbiology Reference Laboratory, NHS GG&C, Glasgow G4 0SF, UK; andrew.smith@glasgow.ac.uk; 40College of Medical, Veterinary & Life Sciences, Glasgow Dental Hospital & School, University of Glasgow, Glasgow G2 3JZ, UK; 41Infectious Disease Epidemiology and Prevention, Statens Serum Institut, DK-2300 Copenhagen S, Denmark; pvb@ssi.dk; 42Department of Public Health and Epidemiology, Faculty of Medicine, Universidad de Los Andes, 12455 Santiago, Chile; mtvalenzuela@uandes.cl; 43National Reference Center for Streptococci, Department of Medical Microbiology, University Hospital RWTH Aachen, 52074 Aachen, Germany; mlinden@ukaachen.de; 44Medical Microbiology and Infection Prevention, Netherlands Reference Laboratory for Bacterial Meningitis, Amsterdam University Medical Centers, Location AMC, University of Amsterdam, 1105 AZ Amsterdam, The Netherlands; n.m.vansorge@amsterdamumc.nl; 45National Reference Centre for Pneumococci, Centre Hospitalier Intercommunal de Créteil, 94000 Créteil, France; Emmanuelle.Varon@chicreteil.fr; 46Department of Infection Control and Vaccine, Norwegian Institute of Public Health, 0456 Oslo, Norway; Brita.Askeland.Winje@fhi.no; 47Department of Pediatrics, Yale New Haven Children’s Hospital, New Haven, CT 06504, USA; inci.yildirim@yale.edu; 48Servicio de Bacteriología Clínica, Departamento de Bacteriología, INEI—ANLIS “Dr. Carlos G. Malbrán”, C1282 AFF Buenos Aires, Argentina; jzintgraff@anlis.gob.ar

**Keywords:** pneumococcal meningitis, serotype distribution, PCV impact, global, meta-analysis

## Abstract

Pneumococcal conjugate vaccine (PCV) introduction has reduced pneumococcal meningitis incidence. The Pneumococcal Serotype Replacement and Distribution Estimation (PSERENADE) project described the serotype distribution of remaining pneumococcal meningitis in countries using PCV10/13 for least 5–7 years with primary series uptake above 70%. The distribution was estimated using a multinomial Dirichlet regression model, stratified by PCV product and age. In PCV10-using sites (*N* = 8; cases = 1141), PCV10 types caused 5% of cases <5 years of age and 15% among ≥5 years; the top serotypes were 19A, 6C, and 3, together causing 42% of cases <5 years and 37% ≥5 years. In PCV13-using sites (*N* = 32; cases = 4503), PCV13 types caused 14% in <5 and 26% in ≥5 years; 4% and 13%, respectively, were serotype 3. Among the top serotypes are five (15BC, 8, 12F, 10A, and 22F) included in higher-valency PCVs under evaluation. Other top serotypes (24F, 23B, and 23A) are not in any known investigational product. In countries with mature vaccination programs, the proportion of pneumococcal meningitis caused by vaccine-in-use serotypes is lower (≤26% across all ages) than pre-PCV (≥70% in children). Higher-valency PCVs under evaluation target over half of remaining pneumococcal meningitis cases, but questions remain regarding generalizability to the African meningitis belt where additional data are needed.

## 1. Introduction

Pneumococcal meningitis is a major cause of childhood morbidity and mortality globally, estimated to have caused 83,900 cases and 37,900 deaths in 2015 [1]. Meningitis is estimated to make up approximately 2% of all severe pneumococcal disease and 12% of pneumococcal deaths [1]. Prior to the introduction of pneumococcal conjugate vaccines (PCV) into routine childhood immunization programs, over 70% of invasive pneumococcal disease (IPD), a serious form of pneumococcal disease that includes bacteremic pneumonia, meningitis, and sepsis, was estimated to have been caused by serotypes targeted by the vaccines currently available [2]. Since then, PCVs have been introduced into infant immunization programs in over 140 countries [3].

Immunizing children with PCV is an effective method for preventing IPD, providing not only direct protection in vaccinated children but also indirect protection (i.e., herd immunity) among unvaccinated individuals by decreasing the circulation of pneumococci of the serotypes included in the vaccines [4,5,6,7]. PCVs currently in wide use include a 10-valent vaccine (PCV10; GlaxoSmithKline (GSK), Synflorix) and a 13-valent vaccine (PCV13; Pfizer, Prevnar13/Prevenar13). Another 10-valent vaccine (Serum Institute of India (SII), Pneumosil) became available in 2019. The 23-valent pneumococcal polysaccharide vaccine (PPV23; Merck, Pneumovax23) is recommended in many countries for older adults and those at high risk for pneumococcal disease but is not widely used [8]. 

Significant reductions in vaccine-type IPD of 41–97%, including pneumococcal meningitis, have been observed in the pediatric population and other age groups after the introduction of PCVs [5,7]. However, these changes are not immediate; indirect protection in unvaccinated individuals takes more time, and several countries in the African meningitis belt had pneumococcal meningitis outbreaks due to vaccine serotypes after the introduction of PCVs [9]. Although low immunization coverage is a possible explanation, it raised questions about the speed and degree of indirect protection in high burden settings without a booster dose, primarily administered in the second year of life. PCV formulations covering 15–24 serotypes have been developed, though they are not yet licensed [10,11,12,13]; these may offer a solution to address much of the remaining disease, but the preventable fraction depends on how much of the remaining disease is caused by the added serotypes. A World Health Organization (WHO) roadmap to defeat meningitis by 2030 was recently endorsed by the World Health Assembly and includes a path to address the remaining leading causes of acute bacterial meningitis, including pneumococcus [14].

Since many countries have now used PCV10/13 extensively, it is possible to examine if serotypes covered by these vaccines have been eliminated in all age groups and what proportion of the remaining disease is caused by the serotypes included in higher-valency PCV formulations under development and in PPV23. We aimed to estimate the global serotype distribution of pneumococcal meningitis cases, by PCV product used and age group, in countries with well-established PCV10/13 routine infant immunization programs and high uptake. This is part of a larger effort investigating the impact of PCV on IPD incidence and serotype distribution to inform current global and national pneumococcal vaccination policies.

## 2. Materials and Methods

### 2.1. Site Identification and Eligibility

Site identification and data collection methods are described in detail elsewhere [15]. Briefly, various methods were used to identify countries conducting serotype-specific IPD surveillance where PCV10 (referring throughout to the GSK product unless otherwise specified, as SII’s vaccine was not in use) or PCV13 was universally recommended for all infants for at least one year by 2018. Known surveillance networks were contacted, and the WHO and experts in the field provided contacts for possible data sources; previous systematic reviews were used to identify potential sites and validate search terms for a literature review that included articles published between 1 January 2011 and 20 December 2018; and International Symposium on Pneumococci and Pneumococcal Diseases (ISPPD) abstracts were reviewed from 2012 to 2018. Individuals at each institution that collected IPD data, including research groups as well as national laboratory testing centers, were invited to participate. All datasets underwent extensive data quality checks to identify any sources of potential biases that could impact the serotype distribution, and these were reviewed with site investigators [15]. 

Assessing eligibility for the meningitis serotype distribution analysis involved a multi-step process: sites had to first have eligible IPD data (step 1), then those sites had to have eligible meningitis data (step 2), which were then assessed to determine the number of years of PCV10/13 use until the serotype distribution stabilized (step 3), and then sites with data after that threshold were included in analyses (step 4). 

Step 1: Data collection eligibility criteria were established to capture years of data where the serotype distribution had likely begun to stabilize, and where there was sufficient serotype data to estimate an unbiased distribution. A site or network had to report serotype-specific IPD case counts, regardless of syndrome. IPD was defined as *Streptococcus pneumoniae* isolated by culture from any normally sterile fluid or using *lytA*-based PCR or antigen-based tests in cerebrospinal fluid (CSF) or pleural fluid. Sites had to have a minimum of four years of post-PCV10/13 introduction surveillance data, including the year of introduction, with a minimum of 12 months of continuous surveillance; have at least 50% of isolates serotyped; have no major changes or biases in surveillance that would affect estimates of serotype-specific percentages; and not be limited to HIV-positive or immunocompromised populations. The year of introduction was defined as the year PCV10/13 was introduced if it was introduced in the first three quarters of the year, or as the following year otherwise. For data submitted in epidemiologic years rather than calendar years, the introduction year was defined accordingly. 

Step 2: Sites had to identify which IPD cases were either confirmed positive for pneumococcus in CSF (CSF+) or had meningitis described as the clinical syndrome in a patient for whom pneumococcus was isolated in blood.

Step 3: Data from sites meeting the above criteria were used to assess the number of years after PCV10/13 introduction needed until the serotype distribution stabilized. To determine when this occurred, the change over time in the annual serotype distribution of all IPD was examined at each site, separately for children and adults. The change in percentage due to individual serotypes, both vaccine types and non-vaccine types, were examined. Particular attention was given to sites with robust data and high-quality surveillance systems. “Stabilization” was defined when trends in vaccine type and prevalent non-vaccine type serotype percentages over time were no longer evident, and the period after this was defined as the “mature” PCV10/13 period. For children under 5 years of age, the number of years of continuous and exclusive PCV10 or PCV13 use required to reach the mature PCV10/13 period varied depending on (a) whether and how long PCV7 was used prior to PCV10/13 introduction (or if there was a period of use of the alternate PCV10/13 product), and (b) whether PCV10/13 was introduced with a catch-up program. For sites without catch-up or prior use of another PCV product, time to reach the mature period was seven years of continuous and exclusive PCV10 or PCV13 use, including the year of introduction. For sites with a PCV10/13 catch-up program, time to reach the mature period was six years and for sites that used another PCV product for three or more years prior to PCV10/13, it was five years. For older children and adults, it took seven years of PCV10/13 use in infants to reach the mature PCV10/13 period, regardless of prior PCV7 use or catch-up, as these had no meaningful observed impact on time to stabilization in these age groups. This process and the thresholds defined here were reviewed by the PSERENADE Technical Advisory Group and the site investigators. 

Step 4: Sites were included in the analyses if they had serotyped meningitis cases during the defined mature PCV10/13 period for their site. Only mature years where the average proportion of children immunized was greater than 70% in the three years preceding were included. WHO-UNICEF estimates of PCV10/13 uptake [16] were used for sites without local immunization coverage information. Sites with concurrent use of PCV10 and PCV13 or that switched between PCV10 and PCV13 and did not use either one for long enough to meet inclusion criteria were excluded. Cases with unknown age were excluded from analyses. Cases from all eligible mature period years were pooled for each site by age group.

The primary analysis was restricted to cases with *S. pneumoniae* identified from CSF (CSF+), given limited availability of clinical syndrome data and differences in the definitions of clinical meningitis across sites (Appendix A). A sensitivity analysis included additional clinically defined meningitis cases (i.e., blood-culture positive, CSF-negative/not tested cases). Additionally, the serotype distribution of CSF+ cases was compared to those with only a clinical meningitis diagnosis within sites with large sample sizes to assess any differences.

### 2.2. Defining Serotype Categories

Cases were grouped into serotype categories for serotypes included in PCV7 (4, 6B, 9V, 14, 18C, 19F, and 23F), PCV10 (PCV7 plus 1, 5, and 7F), PCV13 (PCV10 plus 3, 6A, and 19A), PCV15 (PCV13 plus 22F and 33F) [17], PCV20 (PCV15 plus 8, 10A, 11A, 12F, and 15BC) [18], PCV24 (PCV20 plus 2, 9N, 17F, and 20) [12,13], and PPV23 (PCV24 minus 6A). Serotypes 15B and 15C were grouped as 15BC because they can switch due to a slipped strand mispairing of a tandem thymine–adenine repeat [19]. Non-vaccine type serotypes were defined as serotypes not in the indicated vaccine. 

Serotyping methods for each site are summarized in Appendix A. Cases without a specific serotype identified were rare and were grouped into four categories: “not serotyped”, “untypeable”, “typed, serotype not identified”, and “serogrouped only”. “Not serotyped” cases, those for whom serotyping was not attempted for any reason, were excluded from analyses after site investigators confirmed these to be missing at random. “Untypeable” cases had a comprehensive serotyping methodology performed but did not identify any serotype, such as non-encapsulated strain-prohibiting serotyping, an isolate that produced less capsule under lab conditions and could not be typed phenotypically, or a new serotype; these were grouped with non-vaccine type serotypes and excluded from serotype-specific analyses. “Typed, serotype not identified” cases had serotyping performed with a method that does not assess all serotypes, such as PCR assessing only 37 serotypes; these were grouped with non-PCV10, non-PCV13, and/or non-PPV23 cases depending on the serotyping method used by the site and were excluded from serotype-specific analyses. “Serogrouped only” cases (e.g., 6A/6B/6C/6D), undistinguished cases (e.g., 6A/6C), cases with two serotypes reported, and Quellung Pool-only cases were grouped into serotype categories where possible and excluded from serotype-specific analyses, though these were few.

### 2.3. Analytic Model

The predicted probability of pneumococcal meningitis due to serotype categories and specific serotypes was estimated using multinomial Dirichlet regression [20]. When data were insufficient for the model to converge, distributions were estimated by pooling data across sites. This model assumes that each site has an underlying unknown serotype distribution that varies in its deviation from the “site-averaged” distribution. The model estimates the magnitude of this deviation for all sites as a measure of possible heterogeneity among sites, which is then used along with sample size to determine each site’s weight. When a high degree of heterogeneity exists across sites, sites are weighted more similarly; otherwise, sites are weighted more proportionally to their sample size, so larger sites are weighted more. The model ensures that all proportions in an estimated distribution sum to 1.0. 

The serotype distribution for all observed serotypes could not be modeled because the model cannot estimate distributions when sites have many serotypes with zero counts. To identify which serotypes appeared frequently enough to be modeled, the data were first pooled across sites to estimate the rank of serotypes by product and age group. The top 25 ranking serotypes were selected for each age group and product stratum plus serotype 1, which was added to all strata because it is a key serotype of interest. These serotypes were then used to generate a modeled distribution for each age group and product stratum. 

The robustness of the model is affected by the number of categories (i.e., serotypes) estimated, such that the more categories there are, the less robust the estimates are. Therefore, a hierarchal, or stepped, approach was used where an initial distribution was estimated for serotypes grouped into categories (e.g., PCV13-type), and subsequent models were run on further subdivided categories (e.g., PCV10-type, 19A, 6A, and 3) until reaching individual serotypes. This enabled sites that did not test for all serotypes to contribute to higher-order categories, even if not for some individual serotypes.

For each distribution estimated, whether for serotype categories or specific serotypes within a category, the model was first run on all eligible data using 30 iterations to estimate initial coefficients. Then, the model was run within a bootstrap with 100 replicates of resampling with replacement, using the initial coefficients estimated and stratified resampling based on the covariate used (e.g., PCV product). Within the bootstrap, the model was limited to a single iteration for each replicate. The means of the bootstrap replicates were used as the estimated distribution. For serotype categories, 95% confidence intervals (95% CI) around the mean values were calculated using adjusted bootstrap percentile (BCa) intervals. Due to limited sample size when estimating the distribution of specific serotypes, jackknife resampling was used in place of bootstrapping where one site was removed at a time for specific serotype estimates. In this case, confidence intervals were calculated using the estimated standard errors. When distributions were estimated via pooling, binomial confidence intervals were used. All analyses were performed in R (R Core Team, 2019), and the model used the VGAM package [20].

## 3. Results

### 3.1. Data Included

Of the 76 sites with IPD data eligible for data collection and that participated in the PSERENADE project, 32 PCV13-using sites and eight PCV10-using sites had serotype-specific pneumococcal meningitis cases in the mature PCV10/13 period eligible for this analysis. Reasons for exclusion included: IPD cases were not disaggregated by CSF+ (*N* = 23), no serotyped meningitis cases in the mature PCV10/13 period were reported (*N* = 11), and concurrent PCV10 and PCV13 use (*N* = 4). The majority of meningitis cases were CSF+ (73.3%, range across sites: 24.1–100%) (Appendix A). Most sites (*N* = 26, 65.0%) were from Europe or North America, and only nine (22.5%) were from low- and middle-income countries (Figure 1). Most PCV13 sites (87.5%) previously used PCV7 compared to only two (25.0%) for PCV10 sites. All but four sites used a booster dose schedule, two of which only report data for children < 5 years. Additional site details and characteristics are described elsewhere [15].

### 3.2. Pneumococcal Meningitis Due to Vaccine-Type Serotypes 

In PCV13-using sites, 14.1% (95% CI: 10.4–16.2%) of the remaining CSF+ pneumococcal meningitis during the mature period in children <5 years of age was PCV13-type; among individuals ≥5 years of age, 25.8% (23.6–27.6%) were PCV13-type (Figure 2 and Appendix A). Serotype 3 was the most common PCV13-type in PCV13-using sites (4.0% and 13.1% in <5 and ≥5 years, respectively). 

For PCV10-using sites, due to the large case counts in Brazil (*n* = 210 among <5 years) relative to the other PCV10-using sites (*n* = 43 total among <5 years), data from Brazil were shown separately from all other PCV10-using sites. Data from PCV10-using sites excluding Brazil could not be modeled due to the small sample sizes and were, therefore, pooled. Among children <5 years, the percent PCV10-type was similar in Brazil (4.8%) and the other PCV10 sites (4.9%; Figure 2). The most common PCV10-type serotype was 7F (2.1%). For cases ≥5 years of age, where sample sizes were larger (Brazil: *n* = 707; other PCV10 sites: *n* = 181), the percent PCV10-type cases was lower in Brazil (7.6%; 5.8–9.8%) than in the other sites (14.9%; 10.1–21.0%). Modeled results for all PCV10 sites combined (i.e., including Brazil) are shown in Appendix A.

The percentage of PCV13-type cases was greater in PCV10-using sites than in PCV13-using sites (29.3–40.0% vs. 14.1% for <5 years and 37.0–38.5% vs. 25.8% for ≥5 years). If serotype 6C is considered a PCV13-type serotype because it has possible cross-protection from 6A [24], the difference in the proportion of PCV13-type increases even more. Differences between PCV10- and PCV13-using sites persist for PCV15-type cases but diminish for PCV20-, PCV24-, and PPV23-type cases, which ranged from 54–69% across all age and PCV-use groups. Results restricted to adults aged ≥50 years and ≥65 years showed no meaningful differences compared to results for those aged ≥5 years (Appendix A).

Across both PCV products and age groups, there was wide heterogeneity in the site-specific percentages for the various vaccine-type groups, with the lowest and highest percentages from sites with very small sample sizes (Figure 3). This heterogeneity was evident within regions and overlapped across regions; no clear regional differences were apparent. For the PCV13 sites where modeled estimates were possible, they were consistently within +/-5% of the median of the site-specific estimates.

### 3.3. Serotype Distribution

Serotype-specific percentages are described for all PCV10-using sites combined (i.e., Brazil plus other sites) because the most common serotypes found in Brazil were similar to those in the other PCV10 sites for both age groups (Appendix A); their data were pooled rather than modeled due to small sample size. Among children <5 years of age, the most common serotypes were those not covered by the vaccines in use (Figure 4 and Appendix A). The three PCV13-related serotypes (19A, 6C, and 3) in cases <5 years at PCV10-using sites totaled 42.1% compared to 7.5% at PCV13-using sites. Otherwise, the next most common serotypes at PCV10-using sites were similar to the most common at PCV13-using sites.

Serotype 19A, a PCV13-type, was the top serotype at PCV10-using sites among children <5 years, causing approximately 24% (95% CI: 18.6–30.7%) of cases, and was also common in cases ≥5 years (14.4% in Brazil and 7.8% in other PCV10-using sites). In contrast, at PCV13-using sites, it was uncommon (2.5% for <5 years of age and 3.1% for ≥5 years). 

The second most common serotype at PCV10-using sites among cases <5 years was serotype 6C (10.3%; 95% CI: 6.8–14.9%), another serotype thought to be preventable by PCV13 because of cross-protection from 6A (6C was 1.0% at PCV13-using sites; Figure 2). The proportion of serotype 6C cases was also greater in cases ≥5 years at PCV10-using sites (approximately 10%) than at PCV13-using sites (2.2%). 

Serotype 3 was the top serotype among cases ≥5 years at both PCV13-using sites (13.1%; 8.8–17.4%) and PCV10-using sites (13.9%; 11.7–16.4) and was consistent across most site specific distributions (Appendix A and Appendix A) with the exception of South Africa (Site 55, *n* = 623), where 12F and 8 were the dominant serotypes (approximately 13% each), and serotype 3 was ranked third (5.9%). For children <5 years, the percentage was lower (4.0 and 7.4% at PCV13- and PCV10-using sites, respectively) and was ranked higher at PCV10-using sites (rank = 3) than PCV13-using sites (rank = 8). 

Several non-PCV13-type serotypes that are included in higher-valency PCV products in development (15BC, 8, 12F, 10A, and 22F) were common in both age groups and PCV settings, cumulatively causing between 15 and 36% of pneumococcal meningitis (Figure 4 and Appendix A). Serotype 22F, which is included in PCV15 was generally ranked lower than the others. Serotype 15BC was a leading non-PCV13-type serotype among children <5 years in both PCV10- and PCV13-using sites (6.6 and 11.5%, respectively) but was less common in cases ≥5 years (2.7 and 3.9%, respectively). An exception for children <5 years was South Africa (Site 55, *n* = 172), where serotype 8 was the dominant serotype (38.4% compared to <6% for all other serotypes). Serotypes in the top 5 of at least one age or PCV group that are not included in PCV15, PCV20, or PCV24 include serotypes 24F, 23B, and 23A, which cumulatively caused between 10 and 12% of pneumococcal meningitis across both PCV settings and age groups. 

### 3.4. Sensitivity Analyses

A sensitivity analysis including all meningitis IPD cases (CSF+ cases and non-CSF+ clinically defined cases) was conducted to ensure restriction to CSF+ only did not bias the selection of sites or cases. These showed no meaningful differences from the primary results restricted to CSF+ cases only (Appendix A and Appendix A). A review of the site-specific serotype distributions comparing CSF+ to clinically-defined meningitis cases also showed no meaningful differences for children <5 years. For cases ≥5 years, serotype 3 was more commonly a top serotype for CSF+ cases than for clinically defined meningitis cases, and serotype 19F was sometimes more prevalent in the clinically defined meningitis cases than in the CSF+ cases. Other sensitivity analyses that excluded sites with fewer than five cases in an age group or restricted to sites with data for both age groups did not result in any meaningful differences (data not shown).

## 4. Discussion

We found that in settings where PCV10 or PCV13 have been used for about seven years with primary series uptake above 70%, the percentage of remaining pneumococcal meningitis due to serotypes covered by the vaccines in use was low: 5.3% in PCV10 sites and 14.1% in PCV13 sites in children <5 years and 15.3 and 25.8%, respectively, in older children and adults. This is a substantial reduction compared to the era before PCVs when 70–88% of IPD cases and 62–72% of meningitis cases were caused by PCV10/13-type serotypes in children <5 years of age, depending on the vaccine and region [2,25]. Serotype 19A, a PCV13-type, was rare (≤3%) at PCV13-using sites but caused almost a quarter of cases ≤5 years at PCV10-using sites. A large fraction of the remaining disease was due to serotypes found in higher-valency PCV products in development. PCV15 covered an additional 36% of cases <5 years excluding PCV10-types at PCV10-using sites, although only an additional 7% at PCV13-using sites excluding PCV13-types. PCV20 and PCV24 covered an additional 49–59% in PCV10-using sites and 43–47% in PCV13-using sites. In older children and adults, the percent of pneumococcal meningitis covered by PPV23 was greater than 62%, suggesting there is much vaccine-preventable pneumococcal meningitis still remaining in older age groups. Our results may not represent the local experience of any one country, and while there was heterogeneity observed in vaccine-type distributions, large deviations from the average estimates were predominantly only at small sites. Heterogeneity for some specific non-vaccine serotypes highlights the importance of continued monitoring of serotypes by national surveillance systems. 

Serotype 3 was uncommon in children <5 years of age, as it was in the era before PCVs (estimated 1.4%) [2], causing 3–8% of cases at PCV10-using sites and 4% at PCV13-using sites. However, serotype 3 was the top ranked serotype among those aged ≥5 years in both PCV10- and PCV13-using sites, causing approximately 13–14% of cases. Although population-level direct effects of PCV13 against serotype 3 are not well understood, this suggests that if PCV13 has some direct effects on serotype 3 disease, the indirect effects are likely limited given the high percentage of cases caused by serotype 3 in adults. This has been previously suggested by De Wals in a review of immunologic and effectiveness evidence who concluded that PCV13 may confer some protection in vaccinated children but that it is likely to be lower than for other vaccine serotypes and short term [26]. A subsequent meta-analysis estimated PCV13 effectiveness against serotype 3 IPD in children to be 51–69% [21]. However, none of the studies estimated effectiveness against serotype 3 meningitis, half the studies used case-control methods to assess effectiveness, results only included data from 12 European and North American sites, and results included data with only 4–6 years of PCV13 use [27,28,29,30,31,32,33]. Studies in the US and UK not included in the meta-analysis showed contrasting results after 7 years of PCV13 use, finding no meaningful change in serotype 3 IPD incidence [22,23]. Another small study from Italy that included only six serotype 3 meningitis cases suggests that PCV13 in children may be effective against serotype 3 sepsis and meningitis but not pneumonia [34]. 

The remaining pneumococcal meningitis in PCV10 sites was largely driven by serotypes 19A and 6C, two PCV13-related serotypes, which together caused 34.7 and 23.4% of disease among those <5 years and ≥5 years, respectively. The potential for 19F in PCV10 to provide cross protection to 19A is not supported by our results, as 19A was found to be the dominant serotype of cases <5 years of age at PCV10-using sites, causing nearly a quarter of meningitis in that age group. Although our results for PCV10-using countries other than Brazil are based on sparse data, 19A was commonly seen across PCV10-using sites. Further evidence will be provided by upcoming serotype distribution analyses of all IPD that will increase the number of PCV10-using sites and number of cases, and by analyses of the change in 19A incidence from the pre-PCV to post-PCV periods. Earlier reviews of incidence-based and other studies of serotype 19A were inconclusive [5] and more recent studies have not found evidence of cross-protection from serotype 19F [35,36,37]. Prior evidence of cross-protection by PCV13 from serotype 6A to 6C is stronger [5,24] and consistent with the small (1–2%) percentage of serotype 6C cases at PCV13-using sites compared to approximately 10% at PCV10-using sites. This suggests that by also protecting against serotype 6C, PCV13 could potentially address up to a quarter of remaining pneumococcal meningitis in PCV10-using countries, if the corresponding replacement disease is small.

Another PCV10 product from SII (Pneumosil) has recently been licensed and is important in low- and middle-income countries for its affordability [38]. SII’s PCV10 includes most of the same serotypes as GSK’s PCV10, but replaces serotypes 4 and 18C with 19A and 6A. Using pre-PCV distribution data from Africa and Asia, we expect SII’s PCV10 to cover roughly the same percentage of disease (72–73%) as GSK’s (72–74%), assuming cross-protection from 6B to 6A, but not from 6A to 6C, which was not estimated in the pre-PCV era. SII’s PCV10 covers slightly less than PCV13 if serotype 3 is excluded (76–77%) [2]. The percentage of pneumococcal meningitis in mature PCV10/13 settings covered by SII’s PCV10 could not be estimated as we could not account for serotypes 4 and 18C that are covered by PCV10/13 but not SII’s PCV10. We can speculate that these may expand to their pre-PCV incidence, and possibly greater with replacement disease. However, assessments of the relative impact of PCV products can only be based on comparisons of the change in incidence over time and should evaluate all pneumococcal syndromes, not just the small portion that are meningitis. Further, vaccine product choice involves many factors beyond epidemiologic settings, including programmatic and financial considerations.

Although higher-valency PCV products in development may not further reduce the remaining burden of serotype 3 disease, they target important non-PCV13-type serotypes responsible for much remaining pneumococcal meningitis. PCV20 (PCV13 + 8, 10A, 11A, 12F, 15BC, 22F, 33F) and PCV24 (PCV20 + 2, 9N, 17F, 20) covered more than half (53–69%) of the remaining pneumococcal meningitis across mature PCV10/13 settings and age groups. PCV15 (PCV13 + 22F, 33F) covers less, 9 and 6% of pneumococcal meningitis cases <5 years and ≥5 years, respectively, in PCV13 sites, and 3% in PCV10 sites among both age groups, but is nearest to being available for children. Merck has submitted applications to the U.S. Food and Drug Administration (FDA) and European Medicines Agency (EMA) for licensure of their PCV15 for all ages [10]. Pfizer’s Biologics License Application (BLA) for use of PCV20 in adults 18 years and older has been accepted by the FDA for priority review with a decision expected in June 2021 [39] and Phase 3 trials in children have begun [40]. There are several PCV24 products in the pipeline, including products from Merck [12] and Affinivax, which are currently in Phase 1 testing [13]. If indirect effects for these products are similar to what has been observed for PCV10/13, their use in children <5 years may have more impact than direct immunization with PPV23 in older age groups. 

An important limitation of this analysis is the lack of robust data from PCV10-using sites, with the exception of Brazil, which contributed 86% of those cases. Only one other country (The Netherlands) had more than 10 cases for children <5 years in the mature PCV10 period. This was due to a lack of data rather than insufficient years using PCV10 or lack of participation in PSERENADE. Of 23 countries using PCV10 exclusively, most (15 of 16) of those eligible for PSERENADE contributed data [15]; only eight met analytic eligibility criteria. A future update of this analysis adding data from countries excluded because they had not yet reached the mature PCV10 period will not greatly improve the data paucity problem because they generally had low annual numbers of cases. However, a forthcoming analysis of all IPD cases (not restricting to meningitis) will have more robust results due to larger sample sizes. 

Another important limitation is the paucity of data from high burden settings using a PCV schedule without a booster dose, particularly the African meningitis belt where pneumococcal meningitis outbreaks occur in all age groups and where serotype 1 is a dominant serotype [41,42,43]. PSERENADE received data for only three meningitis belt countries, all using PCV13: Benin and Cameroon contributed two and three cases, respectively, for children <5 years of age, and The Gambia had no meningitis cases in the mature PCV13 period so could not contribute to analyses. Our findings from non-meningitis belt countries that primarily used a booster dose schedule showed serotype 1 consistently caused less than 1% of disease after 7 years of use, compared to 8% in the pre-PCV era among children <5 years [2]. In other PSERENADE analyses published separately in this issue, serotype 1 IPD incidence declined in all ages by 95% after 6 years of PCV10/13 use in non-meningitis belt countries [44]. The persistence of serotype 1 outbreaks in unvaccinated older children and adults in the meningitis belt despite 3–4 years of PCV10/13 use [41,42,43] may suggest that indirect protection may be lower than for other regions, although these results are from the “early” PCV use period so the full effects of PCV13 may not have occurred. In Burkina Faso and Niger, the percentage of pneumococcal meningitis that was PCV13-type was high, approximately 29–45% for children <5 years, with 4–30% due to serotype 1 [41,42]. For cases ≥5 years in Burkina Faso, Niger, and Ghana, 53–74% were PCV13-type and 30–64% were serotype 1 [41,42,43]. These data include cases occurring during meningitis outbreaks, which are commonly caused by serotype 1. Continued monitoring of the serotype distribution in meningitis belt countries using a 3+0 schedule could provide data to understand whether the speed or degree of PCV impact is lower than for countries using a booster dose, or if results are in fact similar after 7 years of PCV13 use. 

An important consideration when interpreting the percentages and serotype distribution results is that a similar sized percentage in the mature PCV10/13 period and in the pre-PCV period represent largely different disease burdens, as PCV has greatly reduced the overall disease burden (i.e., 5% of 100 cases is a much smaller disease burden than 5% of 1000 cases). Consequently, inferences about disease burden cannot be made by comparing percentage sizes alone across age or PCV product strata. In addition, the percentage of any given serotype is affected by the incidence of the other serotypes, such that even if a serotype’s incidence remains stable, the percentage will increase when another serotype’s decreases. Therefore, an analysis of the change in incidence is needed to assess impact, which will be forthcoming in another PSERENADE analysis for sites with incidence data over time. 

## 5. Conclusions

In countries that have used PCV10 or PCV13 for at least 5–7 years and with high uptake, the percentage of pneumococcal meningitis that was vaccine type was less than 15% among children <5 years of age, which is small when compared to percentages above 70% observed prior to PCV introduction, suggesting that PCV10/13 use greatly reduces the proportion of pneumococcal meningitis due to vaccine-type serotypes. Serotype 19A, a PCV13-type, was the most common serotype found in children at PCV10-using sites but rare at PCV13-using sites. Among older children and adults, the percentage vaccine type was <26%, but over 62% was PPV23-type despite common recommendations for PPV23 use among older adults and those at high risk for pneumococcal disease. Higher-valency PCVs currently under evaluation, particularly PCV20 and PCV24, target over half of the remaining pneumococcal meningitis.

## Figures and Tables

**Figure 1 microorganisms-09-00738-f001:**
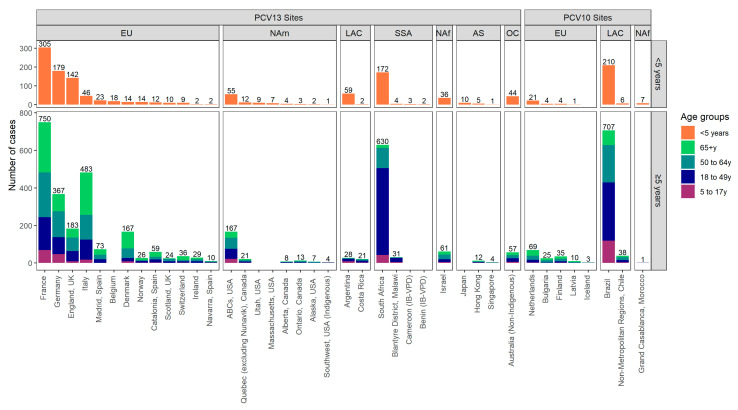
Number of serotyped cerebrospinal fluid positive (CSF+) pneumococcal meningitis cases per site in mature PCV10/13 years by UN region, pneumococcal conjugate vaccine (PCV) product used during years included in the analysis, and age group. Abbreviations: EU = Europe, NAm = North America, LAC = Latin America and Caribbean, SSA = Sub-Saharan Africa, NAf = Northern Africa and Western Asia, AS = Asia, OC = Oceania. PCV13 is Pfizer’s Prevnar13/Prevenar13; PCV10 is GSK’s Synflorix.

**Figure 2 microorganisms-09-00738-f002:**
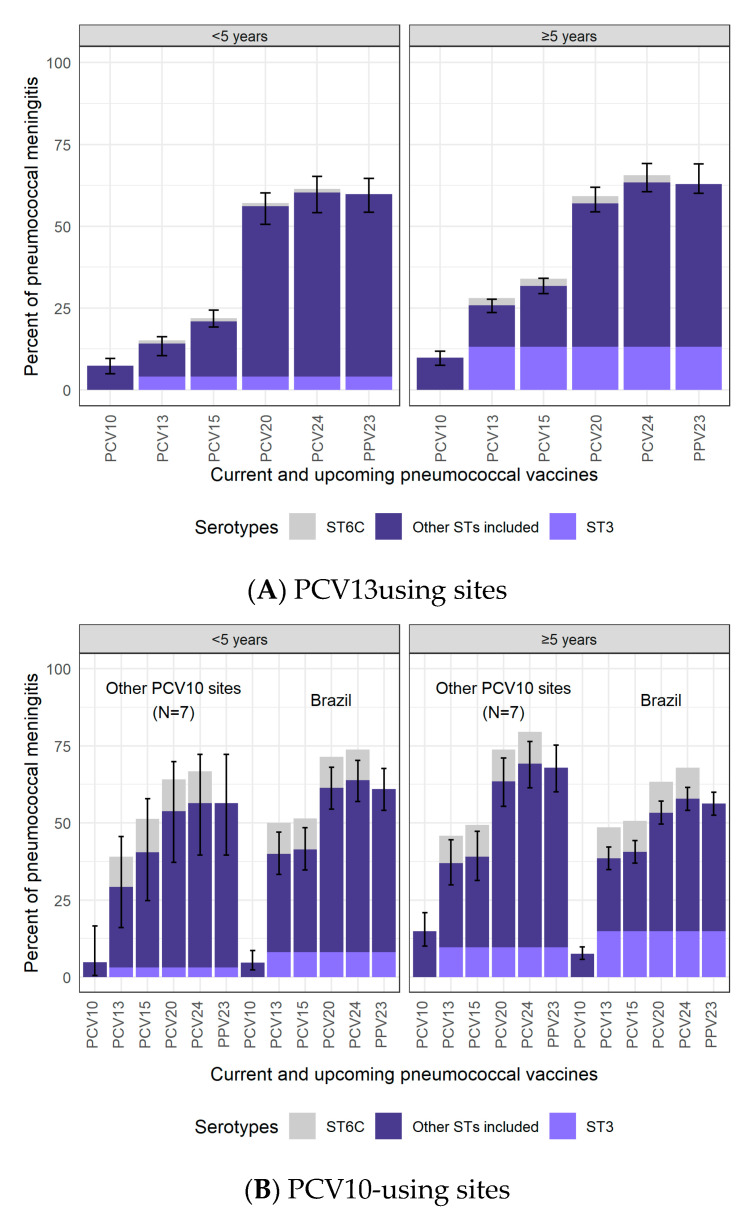
Percentage of CSF+ pneumococcal meningitis cases in the mature PCV10/13 period due to serotypes included in current and upcoming products. PCV13 is Pfizer’s Prevnar13/Prevenar13; PCV10 is GSK’s Synflorix. PCV13 results are modeled output. PCV10 results are a pooled distribution of 210 cases in Brazil and 43 cases in other PCV10 sites for <5 years of age and of 707 cases in Brazil and 181 cases in other PCV10 sites for ≥5 years of age. ST3 is illustrated separately in lighter purple in the bars corresponding to products that include ST3 due to the uncertain effectiveness against ST3 in current products [21,22,23]. ST6C is illustrated in grey above the bars where ST6A is included. Although ST6C is not included in PCV10 or PCV13, PCV13 offers cross-protection through ST6A [24]. ST6A also benefits from cross-protection with ST6B, included in both PCV10 and PCV13. Therefore, ST6A causes a very small fraction of disease in both settings and age groups, and it is not shown. Confidence intervals do not include ST6C, as this serotype is not included in PCV10/13.

**Figure 3 microorganisms-09-00738-f003:**
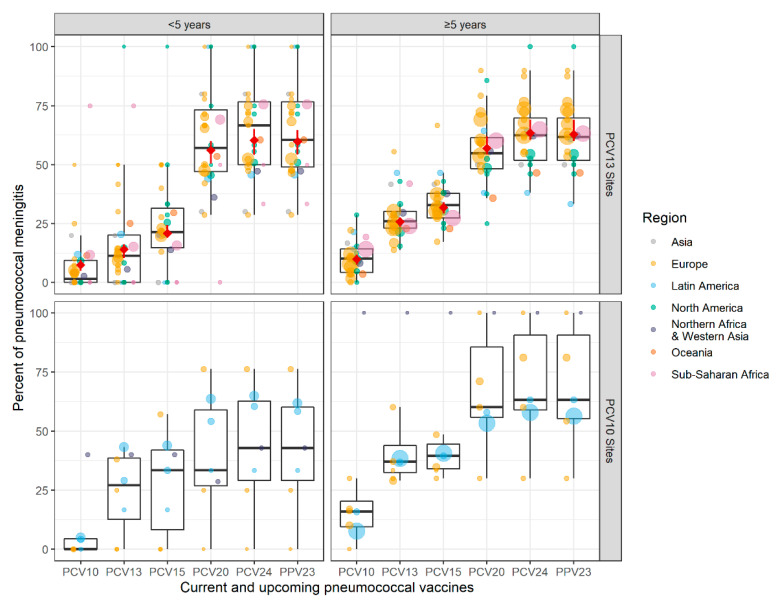
Site-specific percentages of CSF+ pneumococcal meningitis in the mature PCV10/13 period due to serotypes included in current and upcoming products. Each site is represented by a dot, which are colored by region and sized proportionally to the number of cases contributed by that site. The black boxes illustrate the IQR for the site-specific percentages. For PCV13 sites, the modeled results shown in Figure 2 are shown here in red diamonds. Data from Singapore are not shown for confidentiality but contributed to the PCV13 modeled results. PCV13 is Pfizer’s Prevnar13/Prevenar13; PCV10 is GSK’s Synflorix.

**Figure 4 microorganisms-09-00738-f004:**
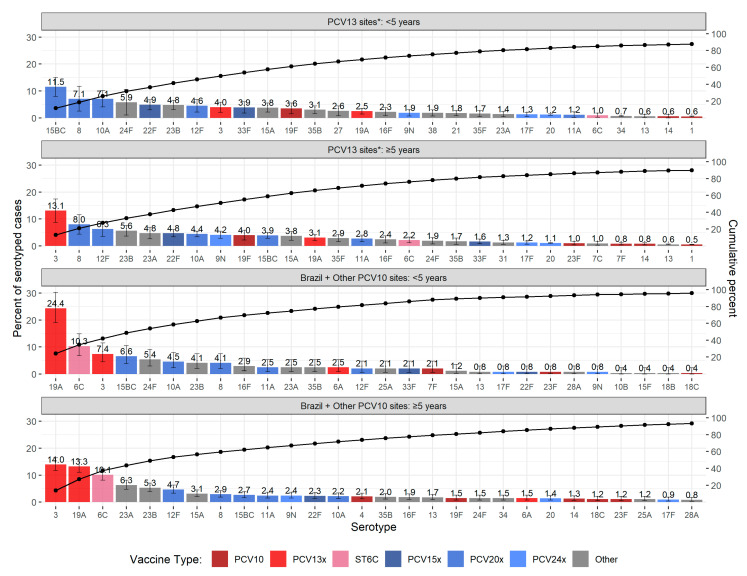
Serotype-specific distribution of CSF+ pneumococcal meningitis in the mature PCV10/13 period. Serotypes are colored by the lowest valency PCV product they are included in. The “x” in the PCV legend represents the extra serotypes included in that product relative to the next lower product (i.e., PCV13x includes serotypes 3, 6A, and 19A not in PCV10). PCV13 is Pfizer’s Prevnar13/Prevenar13; PCV10 is GSK’s Synflorix. Serotype 6C is colored separately because, although it is not included in any product, it is covered through cross-protection with PCV13-type serotype 6A [24]. Morocco and Bulgaria were not included in the PCV10 distribution due to serotyping limitations. * Serotype distribution for PCV13 sites is modeled output for the top 25 serotypes based on a pooled ranking plus serotype 1. Serotype distribution for PCV10 sites is from pooling cases across sites.

## Data Availability

Restrictions apply to the availability of these data. Data were obtained under data sharing agreements from contributing surveillance sites and can only be shared by contributing organizations with their permission.

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
