# Peer review of "Serotype Distribution of Remaining Pneumococcal Meningitis in the Mature PCV10/13 Period: Findings from the PSERENADE Project"

_microorganisms, 2021, doi:10.3390/microorganisms9040738_

Round 1

Reviewer 1 Report

Major comments:

  • My main concern with this analysis is that low/middle-income, higher child-mortality locations are not well-represented, though this is discussed in the Discussion.
  • Clearly describing which results are actual data (pooled) vs. modeled results would be useful, as it’s not always apparent.
  • The Results and Figures are very complex with many dimensions/stratifications, which make them quite difficult to decipher at times. Any possible improvements to simplify or clarify would be helpful.

More detailed comments:

Abstract:

  • Better to say ‘PCV introduction has reduced the incidence of pneumococcal meningitis’?
  • …”with vaccine uptake above 70%” – also, does this refer to the first dose? A complete series? A complete series and a booster, if applicable?
  • Percentages should consistently be presented with 0 or 1 decimal place, preferable 0.
  • “In countries with mature vaccination programs, the proportion of pneumococcal meningitis caused by vaccine-in-use type serotypes is now ≤25% compared to ≥70% in children pre-PCV.” – does this refer to children <5y? If so, confusing because abstract says PCV10-types caused 5% and PCV13-types caused 14.1% in children <5, but here says ≤25%?
  • “Higher-valency PCVs under evaluation target over half of remaining pneumococcal meningitis serotypes.” – would perhaps be informative to say the proportion of remaining pneumococcal meningitis CASES that would be targeted, rather than proportion of remaining serotypes?

Introduction:

  • Line 114: are more updated disease burden estimates available? These are 6 yrs old.
  • Line 120: PCVs

Methods:

  • Line 175: were sites that conduct pneumococcal meningitis surveillance ONLY (not other IPD syndromes) included in analysis?
  • Lines 195-196: “Particular attention was given to sites with robust data and high-quality surveillance systems.” – what does this mean exactly? Were these sites weighted more in analysis? Or these sites were primarily used to define the maturation thresholds?
  • Line 196: How does the concept of stabilization apply in settings that experience pneumococcal outbreaks (e.g. West African countries)?
  • Lines 218-219: how many cases were excluded because of unknown age?
  • Line 240: how often were cases ‘not serotyped’ at random? I would imagine that this would not necessarily be random, but might be cases from lower-performing surveillance sites, or cases with specimens that were not transported to a site with serotyping capability, etc.

Results

  • Line 338: perhaps it’s better to clarify in the Methods that the PCV13 site results are modeled and the PCV10 site results are pooled.
  • Lines 355: “indicating no evidence of regional differences” – maybe a bit too strong, given data limitations. Could perhaps instead say “no clear regional differences were apparent” or something along those lines.
  • Section 3.3 Serotype distribution – I struggled with the logical order of this section/paragraph – it seems quite jumbled to me. Perhaps group these results into shorter paragraphs with clear topic sentences to help orient the reader?
  • Line 372: “for the next seven serotypes, 95% CIs overlapped 5%.” – to me, this isn’t very informative. Could consider giving the range of %s for the next 7 serotypes instead? Or just delete this part of the sentence?
  • Lines 382-383: Site 55 – why is this mentioned, as site numbers haven’t been previously used and don’t seem relevant to the presentation of results.

Discussion:

  • Lines 440-441: “any heterogeneity observed in vaccine-type distributions was predominantly only at small sites” – I’m not sure I agree with this statement. The results do show heterogeneity, it’s just that the analyses are weighted towards larger sites, many of which are Western European. I don’t think there is a need to minimize the observation of heterogeneity, as heterogeneity is to be expected in any global pooled analysis.
  • Lines 481-482: specifically mention the product name (Pneumosil)?
  • Line 516: it’s not really a “lack” of data from meningitis belt countries, but rather “limited” data.
  • The target population for PPV23 is described in a number of different and inconsistent ways, and should be made consistent:
    • Line 129: “older adults and those at high risk for pneumococcal disease”
    • Line 148: “high-risk children and adults”
    • Line 437: “older children and adults”
    • Line 554: “elderly and high risk adults”

Figures

  • Figure 2:
    • After reading the figure legend, I understand why serotypes 6C and 3 are presented separately. However, since this complicates the figure, is not immediately intuitive, and is not at all mentioned in the results, I’m wondering if this level of detail can be either left out of the figure and perhaps just be presented as a supplemental figure? Or if this is important to include in the main manuscript, perhaps it should be mentioned in the Results text, along with an explanation of the significance.
    • Are these modeled results (as suggested by the legend for Figure 3)? If so, that should be clearly indicated in the Figure title.
  • Figure 3:
    • “For PCV13 sites, the modeled results shown in Figure 2 are shown here in red diamonds.” – why this wouldn’t be done similarly for pooled results from PCV10 sites?
    • “Data from Singapore are not shown for confidentiality but contributed to the PCV13 modeled results.” Why is Singapore specifically called out here, when other sites had low case counts as well?
  • Figure 4:
    • I don’t understand why the “x” notation is needed, if you already indicated that ‘Serotypes are colored by the lowest valency PCV product that they are included in’ – doesn’t that cover it?
    • “Morocco and Bulgaria were not included in the PCV10 distribution due to serotyping limitations.” – what does this mean? Are these countries included in other analyses?
    • Why is the label “Brazil + Other PCV10 sites” used here? Why not just say “PCV10 sites”, as in Figures 1 & 3?
    • The last sentence of the legend appears incomplete

Supplemental material

  • Generally, the supplemental tables and figures should be numbered in the order they are referred to in the next.
  • Supp Figure 5: why are site numbers used here, instead of names? The numbers are not informative, and not used elsewhere.

Reviewer 2 Report

The authors present a critical analysis of the pneumococcal serotypes associated with meningitis in regions with mature PCV programmes. The data presented here will be useful to public health experts, policy makers and decision makers globally. 

Major comments
1.    The authors have put a lot of thought on the inclusion criteria for datasets and countries. However, some of the decisions while they seem reasonable, are not supported with a rationale or justification in this paper. 
2.    There is significant scarcity of data for sites using PCV10 (apart from Brazil), especially for children under 5 years old. The authors have chosen to combine the analyses of the data from the other sites, however, given the potential heterogeneity across sites demonstrated by the authors, I am not convinced that this is the right approach and question the conclusions drawn. The authors should consider excluding sites where the data are scarce. 
3.    I find it concerning that there were only 2 and 3 cases from Benin and Cameroon respectively from hospital surveillance among under 5s. This is unlikely representative of the actual burden of pneumococcal meningitis but rather a reflection of weak surveillance systems and limited laboratory capacity to identify and serotype pneumococci. Furthermore, these data are limited to under 5s. Hence, given these limitations, it would seem reasonable to exclude these data and acknowledge the data gaps. 
4.    The authors collated data from different countries collected using various methodologies ranging from hospital surveillance to population-based or national surveillance. The data collection method invariably influences the total number of cases detected and the serotype distribution observed i.e. sentinel surveillance in under 5s only vs. population-based surveillance. Hence, some of the validity of some of the comparisons made and conclusions could be limited. The authors should consider these factors further and discuss these limitations further. 

5. Further discussion of the limitations of this study is required.  

Minor comments
Lines 170-180: Please provide the rationale for four years of post-PCV surveillance and the need to have 50% of isolates serotyped.
Line 212: Should be “included in the analyses”
Lines 322-324: No need to include the serotypes in each vaccine in the Figure 2 legend as these are already described in the methods section.  
Line 326: Please check “for in Brazil”
Lines 332-334: The cross protection information is necessary in the figure legend
Line 334: Why do the confidence intervals not include serotype 6C?
Lines 335-343: The authors decided to separate the Brazil analyses for PCV10 sites because of the large number of cases from Brazil. However, that left 43  cases for under 5s from everywhere else. How representative are these data?  
Figure 3: The data presented in this figure would be much easier to follow and interpret in a table or some other format that would allow your audience to identify data from specific countries within the different regions across the various age groups. Colours for North and South America are very difficult to differentiate.  

Reviewer 3 Report

Serotype Distribution of Remaining Pneumococcal Meningitis in the Mature PCV10/13 Period: Findings from the PSERENADE Project

This is an interesting, well-thought and well-written manuscript in which the authors study the serotypes distribution in pneumococcal meningitis in PCV period. This work is very well planned and contributes much information to the epidemiology of this pathogen. This manuscript should be suitable for publication in Microorganisms after addressing several minor points:

  1. Lines 100, 345, 348… should be “vs”

  2. How would Figure 1 be affected if Madrid, Catalonia and Navarre were unified as Spain? And the same for Canada (Quebec, Alberta, Ontario) and UK. Clarify it, please

  3. Line 450. Add De Miguel et al., 2020. CID to the reference. Vaccination with PCV13 in adults seems to control IPD cases by PCV13 serotypes including serotype 3. Introduction of PCV13 or PPV23 in the adult calendar of certain Spanish regions reduced the IPD cases by PCV13 serotypes by up to 25% and 11%, respectively, showing a decrease of serotype 3 when PCV13 was used.
  4. Supplementary Table 4, explain PCR35/PCR38 ¿real-time polymerase chain reaction or conventional?
